# An Array of Flag-Type Triboelectric Nanogenerators for Harvesting Wind Energy

**DOI:** 10.3390/nano12040721

**Published:** 2022-02-21

**Authors:** Zhiqiang Zhao, Bin Wei, Yan Wang, Xili Huang, Bo Li, Fang Lin, Long Ma, Qianxi Zhang, Yongjiu Zou, Fang Yang, Hongchen Pang, Jin Xu, Xinxiang Pan

**Affiliations:** 1Maritime College, School of Electronics and Information Technology, Guangdong Ocean University, Zhanjiang 524088, China; zhaozhiqiang929@gdou.edu.cn (Z.Z.); weibin2@stu.gdou.edu.cn (B.W.); 2112010010@stu.gdou.edu.cn (X.H.); boli@gdou.edu.cn (B.L.); linfang@gdou.edu.cn (F.L.); malong@gdou.edu.cn (L.M.); zhangqx@gdou.edu.cn (Q.Z.); neomailyf@gdou.edu.cn (F.Y.); 2Guangdong Provincial Shipping Intelligence and Safety Engineering Technology Research Center, Zhanjiang 524088, China; 3Southern Marine Science and Engineering Guangdong Laboratory (Zhanjiang), Zhanjiang 524006, China; 4Marine Engineering College, Dalian Maritime University, Dalian 116026, China; e0701746@u.nus.edu (Y.W.); zouyj0421@dlmu.edu.cn (Y.Z.)

**Keywords:** flag-type, triboelectric nanogenerator, wind energy, array, network generation

## Abstract

Harvesting wind energy from the ambient environment is a feasible method for powering wireless sensors and wireless transmission equipment. Triboelectric nanogenerators (TENGs) have proven to be a stable and promising technology for harvesting ambient wind energy. This study explores a new method for the performance enhancement and practical application of TENGs. An array of flag-type triboelectric nanogenerators (F-TENGs) for harvesting wind energy is proposed. An F-TENG consists of one piece of polytetrafluoroethylene (PTFE) membrane, which has two carbon-coated polyethylene terephthalate (PET) membranes on either side with their edges sealed. The PTFE was pre-ground to increase the initial charge on the surface and to enhance the effective contact area by improving the surface roughness, thus achieving a significant improvement in the output performance. The vertical and horizontal arrays of F-TENGs significantly improved the power output performance. The optimal power output performance was achieved when the vertical parallel distance was approximately 4*D*/15 (see the main text for the meaning of *D*), and the horizontal parallel distance was approximately 2*D*. We found that the peak output voltage and current of a single flag-type TENG of constant size were increased by 255% and 344%, respectively, reaching values of 64 V and 8 μA, respectively.

## 1. Introduction

In the era of the Internet of Things (IoT), the widespread implementation of low-power wireless sensors, portable electronics, and wireless transmission devices is required. This requirement raised concerns regarding the associated increase in demand for energy supply, owing to the large number and dispersed locations of the required devices.

Wind power is a promising renewable energy source for wireless sensors and wireless transmission equipment because it is one of the cleanest sources of energy, with a wide distribution and high availability. Currently, wind power is mostly generated using large-scale equipment containing horizontal-axis propeller wind turbines with runners to drive generators for creating electricity [1,2]. A minority of wind power generation methods implement miniaturized equipment. Li et al. introduced a blade structure that uses the piezoelectric effect to collect wind energy. After changing the traditional arrangement of a flutter device parallel to the flow direction to a suspended cross-flow arrangement, the vibration was amplified by an order of magnitude [3]. Perković et al. used the Gnus effect to generate electricity using high-altitude wind energy [4]. Ji et al. also used piezoelectric principles to fabricate a small wind energy collection device in a resonant cavity, which greatly improved the conversion efficiency compared to other small wind energy collection devices [5]. Perez et al. used a combination of the tremor effect and electret conversion to convert the flow-induced motion of a film into electrical energy [6]. To date, although the collection of wind energy is gradually becoming diversified, various problems remain concerning the implementation of miniaturized wind energy collection devices, such as their complex structure and low efficiency.

Triboelectric nanogenerators (TENGs) possess significant advantages over electromagnetic and piezoelectric power generators. Much work has been conducted on TENGs for wind energy harvesting. Various studies proposed a new type of power generation technology, which couples electrostatic induction with the triboelectric effect to efficiently convert mechanical energy into electrical energy [7,8,9,10,11,12,13]. In the field of wind energy collection, a substantial number of wind energy collection devices based on TENG were designed. For example, Yang et al. used a combination of fluorinated ethylene propylene (FEP) film and aluminum film in an acrylic tube to produce a self-powered sensor system that integrates wind energy collection and vector wind speed measurement [14]. Bae et al. used gold as a conductive material and used rigid plates coupled with flexible fabrics to generate electricity. This device could generate an instantaneous voltage of 220 V and a short-circuit current of 60 μA at a wind speed of 15 m/s, with an average power density of 0.86 mW [15]. Wang et al. reported on a systematic study of a combination of polyimide film and a TENG driven by elasto-aerodynamics, composed of copper electrodes, and fixed in an acrylic fluid channel, and derived a relationship between the output power density and the size of the fluid channel [16]. Zhao et al. proposed a flag-shaped TENG based on a weaving method, which collected high-altitude wind energy using power-sensing equipment to charge a lithium battery [17]. Wang et al. designed a sandwich flag-type triboelectric nanogenerator (F-TENG), which has a simpler structure and lower cost than other wind energy collection devices and maintained good output performance under conditions of high humidity. [18]. In addition, some scholars proposed the “all-in-one package” technology containing both TENG and PNG, which have great potential in harvesting energy from water waves (blue energy), airflow (wind energy), sound frequency (acoustic energy), vibrations/mechanical motions (such as body motions activities/sleeping), and in vivo body motions. These harvesting technologies are expected to dominate the future smart world [19,20,21,22].

Published research on wind energy harvesting based on TENGs precluded the full use of the characteristics of polytetrafluoroethylene (PTFE) electrets and aerodynamics, resulting in a low power output. This study explores a method for collecting wind energy based on a TENG providing direct power to low-power-consumption wireless terminals to solve the problems of energy needs, scattered arrangement, and environmental pollution. The optimal array structure of an F-TENG based on wind-induced fluid vortex-induced vibration is also discussed.

## 2. Materials and Methods

### 2.1. The Working Principle of a Single F-TENG

According to the power generation principle of an F-TENG, each contact surface of the triboelectric pair is regarded as a node and simplified to an equivalent capacitance model [23]. Based on the principle of the contact-mode freestanding TENG, the intrinsic output performance of the TENG can be deduced using electrodynamics. The formulas for calculating the open-circuit voltage *V*_OC_ and the short-circuit charge transfer *Q*_SC_ are as follows:(1)$$V_{\mathrm{OC}}=\frac{2\sigma x(t)}{\varepsilon_{0}};$$
(2)$$Q_{\mathrm{SC}}=\frac{2S\sigma x(t)}{d_{0}+g}.$$

In these equations, the ratio of the thickness of the dielectric material between the two electrodes to the relative dielectric constant is defined as the effective thickness constant *d*_0_, and the electrode gap *g*, whereas *ε*_0_ is the relative dielectric constant of air, and *x*(*t*) is the distance between the two triboelectric surfaces, which changes with time according to the external force-triggered condition. The quantity *σ* is the surface charge density of the PTFE and the conductive ink film, whereas *S* is the actual contact area between the PTFE and the conductive ink film. The relationship between the voltage (*V*), the amount of charge (*Q*), and the distance between the two triboelectric charges (*x*) of the F-TENG can be deduced using the electrodynamic theory [24]:(3)$$V=-\frac{Q}{C}+V_{\mathrm{OC}}=-\left(\frac{d_{0}+g}{\varepsilon_{0}S}\right)Q+\frac{2\sigma x(t)}{\varepsilon_{0}}.$$

Based on Equation (1), an increase in the surface charge density of the PTFE effectively increases the output of the F-TENG charge. PTFE is an electret material [25] and its charge can be retained on its surface for a long time. Three F-TENGs of the same size were fabricated (length *D* = 150 mm, breadth *B* = 75 mm, thickness *T* = 80 μm). The PTFE films were treated with different grinding procedures.

The working principle of the F-TENG is shown in Figure 1a. Owing to the different stiffnesses, the bending degrees of the PTFE and the electrode are different, and flutter deformation occurs under the action of wind, resulting in alternating contact and separation between the PTFE and the electrode. In the F-TENG’s original state, the electrode and PTFE are separated, and there is no charge transfer. When the F-TENG is bent upward, the PTFE comes into contact with the lower electrode. Owing to the difference in the electron affinity between the PTFE and electrode, the electrode and the surface of the PTFE carry positive and negative charges, respectively. As the electrodes are separated, a potential difference is created between the two electrodes, driving electrons through an external circuit from the upper electrode to the lower electrode and generating a current. When the F-TENG is bent downward, the PTFE is in contact with the upper electrode, which is positively charged. If the PTFE is separated from the upper electrode, the electrons flow in the opposite direction, and an opposite current will be generated in the external circuit. Therefore, owing to the alternating contact and separation between the PTFE and the conductive ink film electrode, a short-circuit alternating current is formed as the charge flows repeatedly between the two layers of the conductive ink film electrode.

The COMSOL Multiphysics 5.6 software package was used to simulate the electrical distribution of the F-TENG during flutter deformation, as shown in Figure 1b. In the initial state, assuming that the surface of the PTFE carried a negative charge that was uniformly distributed, each of the two electrodes carried an equal amount of positive charge, and the total amount was equal to the negative charge mentioned above. As shown in Figure 1b(i), when the PTFE film is located at the center of the electrodes on both sides, the electric potential on the respective electrodes is equal, without any potential difference. When the lower electrodes of the PTFE film are close to each other (as shown in Figure 1b(ii)), a potential difference is induced between the lower and upper electrodes. As shown in Figure 1b(iv), the PTFE then moves in the upward direction until it contacts the upper electrode, generating an electromotive force opposite to that in Figure 1b(ii).

### 2.2. The Fabrication and Pre-Grinding of an F-TENG

The structure of the designed TENG is illustrated in Figure 2a. A PTFE film with a thickness of 30 μm was chosen as the substrate material owing to its low weight, good workability, and electron affinity. Using screen-printing technology, conductive ink, as the flexible electrode, was coated onto a polyethylene terephthalate (PET) substrate with a thickness of 25 μm. The F-TENG adopted a “sandwich flexible structure” in the sequence electrode-PTFE-electrode, wherein an electrode was employed as both a triboelectric material and a conductive material. The surface microstructures of the PTFE films of the three respective types of F-TENGs we fabricated are shown in Figure 2b. The roughnesses of the three types of PTFE films obtained by grinding with different sandpapers (untreated group, ground by P10000 sandpaper, and ground by P400 sandpaper, the normal pressure/force is about 2 N and horizontal speed of grinding with the sandpaper is about 0.05 m/s) are obviously different.

### 2.3. Experimental Setup

The experimental setup for testing the F-TENGs’ output performance in a low-speed wind tunnel is shown in Figure 3. The dimensions of the wind tunnel were 0.25 m (width) × 0.25 m (height) × 1 m (length), and the wind speed was varied between 2 m/s and 13 m/s using an inverter for speed control. A programmable electrometer (Keithley Model 6514, Cleveland, OH, USA) and Data Acquisition Board (NI DAQ, Shanghai Enai Instrument Co., Ltd., Shanghai, China) were used to realize the measurement and acquire the experimental data. A 3D printed structure was used to fix an F-TENG inside the wind tunnel. The position of the F-TENG could be changed in subsequent experiments.

### 2.4. Horizontal and Vertical Parallel Arrays of Double F-TENGs

The special membrane structure of an F-TENG allows for it to be stacked inside a small space to form power generation device arrays. Because of the effect of flow-induced vibration, interference occurs when two flags are positioned horizontally or vertically in parallel, resulting in changes in the vibration amplitude and frequency. With respect to the horizontal parallel arrays, Zhu [26] designed and performed a series of numerical simulations on the interaction of a pair of horizontal parallel flexible flags separated by a dimensionless distance (0 ≤ *D_t_* ≤ 5.5) in a flowing, incompressible viscous fluid using the immersed boundary (IB) method at lower Reynolds numbers (40 ≤ *Re* ≤ 220). Based on the dimensionless IB formulation in component form, the elastic potential energy density *E* associated with the flags, and the Eulerian coordinates, *X* and *Y* (*x* and *y* components, respectively), of the flags, whose associated Lagrangian coordinate is α, are computed as follows:(4)$$E=\frac{1}{2}{\hat{K}}_{s}{\int_{\Gamma }\left(\sqrt{{\left(\frac{\partial X}{\partial \alpha }\right)}^{2}+{\left(\frac{\partial Y}{\partial \alpha }\right)}^{2}}-1\right)}^{2}d\alpha +\frac{1}{2}{\hat{K}}_{b}\int_{\Gamma }\left({\left(\frac{\partial^{2}X}{\partial \alpha^{2}}\right)}^{2}+{\left(\frac{\partial^{2}Y}{\partial \alpha^{2}}\right)}^{2}\right)d\alpha ;$$
(5)$$\frac{\partial X}{\partial t}(\alpha ,t)=\int_{\Omega }u\delta (x-X(\alpha ,t))\delta (y-Y(\alpha ,t))dxdy;\;$$
(6)$$\frac{\partial Y}{\partial t}(\alpha ,t)=\int_{\Omega }v\delta (x-X(\alpha ,t))\delta (y-Y(\alpha ,t))dxdy.$$

In the above equations, *x* and *y* are the Eulerian coordinates associated with the fixed computational domain, *α* is the Lagrangian coordinate associated with the moving flags, and *t* is the time. The quantities *u* and *v* are the components of the fluid velocity along the *x* and *y* axes, respectively. The symbol Γ represents the flags, and Ω represents the two-dimensional fluid domain (the rectangle with an aspect ratio of 2:1). The quantities ${\hat{K}}_{s}$ and ${\hat{K}}_{b}$ are the stretching/compression coefficient and bending modulus of the flags, respectively. The flag position (*X*, *Y*) is updated by applying Equations (5) and (6). Previous studies determined that the distance between the flags is the key factor in determining the energy received by the flag.

With respect to vertical parallel arrays, Wang [27] used the value of the dimensionless parameter *St*, which represents the vibration characteristics of objects, to describe the change in the swing of a single flag as a function of wind speed:(7)St=fA/U.

The peak value of the trailing edge displacement of the flag is represented by *A*, the swing frequency of the trailing edge by *f*, and the inflow velocity by *U*. Under different inlet wind speeds, the two flags may move in four different coupling modes: static, co-directional swing, reverse swing, and transition state. The experimental results laid the experimental foundation for our research on the vertical parallel mode. The four different coupling modes discussed to provide a reference for the optimal layout of the F-TENG in a vertical parallel array, which optimizes the output performance.

## 3. Results and Discussion

### 3.1. The Output Performance of a Single F-TENG with Different Pre-Grinding

As shown in Figure 4, the maximum output voltage (*V**_max_*) obtained experimentally for the untreated group is approximately 15–18 V at a wind speed of 7.2 m/s. The *V_max_* of the F-TENG fabricated by grinding the PTFE with P10000 sandpaper is approximately 22 V. The roughness of the PTFE after grinding with P400 sandpaper is the greatest among the three samples, and the voltage output obtained with this sample of PTFE is also the largest. At a wind speed of 7.2 m/s, the output voltage and current of the F-TENG ground by P400 reaches 64 V and 8 μA, respectively, values that are approximately 255% and 344% higher, respectively, than those of the untreated group.

Three F-TENGs were prepared using different pre-grinding methods. F-TENG #1 and #2 were ground using P10000 sandpaper, and F-TENG #3 was ground using P400 sandpaper. The voltages, currents, transferred charges, and power outputs (*P* = *I*^2^*R*) obtained experimentally are displayed in Figure 5. The variations in the electrical properties of the three F-TENGs all show similar trends, but F-TENG #3 clearly delivered a superior level of performance. The analysis revealed that the electret characteristics of the PTFE film caused an increase in the surface charge density σ after the grinding treatment. According to Equation (1), in the process of alternating contact and separation between the PTFE film and the conductive ink film, the open-circuit voltage *V*_OC_ and the short-circuit charge transfer *Q*_SC_ increase in the same proportion, which leads to an increase in the current and power in the circuit. The micro-nano scratches that formed on the surface of the PTFE film after the pre-grinding treatment enhanced the degree of triboelectric contact between the PTFE and the conductive ink film, and further increased the amount of triboelectric charge. Therefore, the pre-grinding treatment is an important factor in improving the power generation performance of F-TENGs.

### 3.2. Horizontal and Vertical Parallel Array Characterization of the Double F-TENGs

The interaction between the two flags in different modes of parallel motion can lead to different internal triboelectric states in an F-TENG. This characteristic can therefore be used to improve the power generation performance of F-TENGs. The use of an F-TENG to collect wind energy and increase the amplitude of vibration is beneficial for improving the output of electricity [18]. Two F-TENGs were arranged in horizontal parallel and vertical parallel modes, respectively, for interference experiments in a low-speed wind tunnel, with the results shown in Figure 6.

In the vertical parallel mode experiments, F-TENGs #1 and #2 with similar performance outputs were selected and placed in the vertical parallel mode. The vertical distance between the two flagpoles (*e*_P_) was adjusted to 1*D*/15, 2*D*/15, 4*D*/15, 6*D*/15, and 8*D*/15, owing to the limitation of the wind tunnel size, the maximum value that could be obtained for *e*_P_ was 8*D*/15. The power output characteristics at different values of *e*_P_ were measured at a wind speed of 7.2 m/s. The contrast diagram of the maximum total output power when F-TENG #1 and #2 were connected to the circuit in series is shown in Figure 6a. When the interval *e*_P_ = 1*D*/15, the swing phases of the two flags were identical, and the total output power was low. When *e*_P_ = 2*D*/15, the swing phases of the two flags were chaotic, and the output was unstable, indicating that the configuration lay in the zone between phase coherence and anti-phase coherence. When *e*_P_ ≥ 4*D*/15, the swing phases of the two flutter flags were opposite. When *e*_P_ = 4*D*/15, the maximum output power of the array with two F-TENGs was achieved at 67.8 μW, whereas the power density per unit area was 6.03 mW/m^2^. The analysis indicated that the two flags fluttered in the opposite phase and were close to each other, which increased the triboelectric strength and contact area between the PTFE films and the electrodes, thus increasing the output of the F-TENGs. With an increase in the distance between the two flags, the contact area decreased, and the maximum output power decreased and tended to stabilize.

In the horizontal parallel mode experiments, as shown in Figure 6b, F-TENG # 3 was selected as the measurement unit, and an F-TENG of the same size was placed in front of it as the interference source. By changing the horizontal distance between the two flagpoles (*e*_T_), the output characteristics of the rear flag (F-TENG #3) were analyzed to determine the optimal distance for achieving the maximum power output. We assumed that the lengths of the F-TENG were *D* and assumed 0.5*D* as the moving distance. When *e*_T_ = 2*D*, the output power of F-TENG #3 reached its maximum and the maximum output power increased by approximately 17.9% compared to that of the single flag. With an increase in *e*_T_, the interference gradually weakened, and the output power gradually decreased. When *e*_T_ = 2.5*D*, the output power was close to the output power of the single F-TENG. The reason for this phenomenon is that the front flag interfered with the airflow and formed a shedding vortex, which increased the vibration amplitude of the rear flag [26], thus increasing its output power.

### 3.3. Generation Performance of the F-TENGs Network

Based on the aforementioned results, we arranged twelve F-TENGs to form a power generation network, as shown in Figure 7a. In this array, the distance *e*_T_ was fixed at 1*D*. By adjusting the vertical distance *e*_P_ in each group, the performance of the power generation network was measured; limited by the size of the wind tunnel, when three rows of F-TENGs were arranged in a vertically parallel mode, the maximum possible vertical distance *e*_P_ was 6*D*/15. A circuit was designed for wind energy harvesting with an array of LEDs as the power output. The 2 electrodes of an F-TENG were sequentially connected to a rectifier to provide a direct current output, and 12 F-TENGs were connected in multiple modes to a power load composed of 100 LEDs. By setting the wind speed equal to 8.0 m/s in the wind tunnel, 100 LEDs are efficiently lit by the 12 F-TENGs arranged at separations of 4*D*/15 in the vertical direction and at separations of 1*D* in the horizontal direction, as shown in Figure 7a,b.

## 4. Conclusions

The placement of a TENG device is challenging because of its aerodynamic and electrostatic characteristics. It is important to choose a reasonable distance between each F-TENG device to improve the output power. Therefore, a procedure for enhancing the performance of an F-TENG array and the associated networking design was proposed and investigated in the present paper. The designed F-TENG array effectively converted the flutter energy contained in wind into electrical energy by enhancing the triboelectric interaction between the PTFE membrane and a carbon-coated membrane, as well as increasing the charge on the PTFE membrane. In wind tunnel experiments, the maximum output voltage and current of the F-TENG array were enhanced by 255% and 344%, respectively, after pre-grinding the PTFE dielectric membranes. The F-TENG network was assembled in a configuration that enhanced its power generation performance. When the network contained two F-TENGs arranged in horizontal parallel mode at a separation of 1*D*, the peak output power of F-TENG #3 was increased by 17.9%. When F-TENG #1 and #2 were arranged in vertical parallel mode at a separation of 4*D*/15, the peak output power increased by 24.3%, and the power density per unit area reached 6.03 mW/m^2^. When the network contained 12 F-TENGs separated by 1*D* in the horizontal direction and by 6*D*/15 in the vertical direction, under 8.0 m/s wind conditions, the array was able to light up 100 LEDs simultaneously. In this study, to efficiently place more energy harvesting elements in a small space, we selected an optimal F-TENG arrangement scheme. The arrangement was chosen in such a way that each F-TENG element can maximize the energy harvesting efficiency. The array of fabricated F-TENGs developed and tested in this study shows great potential for applications in wind energy harvesting and power supply for wireless sensors and devices.

## 5. Experimental Section

### 5.1. The Fabrication of the F-TENG Unit and Network

The construction process of the F-TENG is shown in Figure 2a. The F-TENG and the flagpole were connected using sticky tape. The size of the F-TENG was 150 mm (length) × 75 mm (width) × 80 μm (thickness). The PTFE and PET membranes were 30 μm and 25 μm in thickness, respectively. A conductive carbon ink electrode with a micrometer thickness was attached to the back side of the PET to serve as the flexible electrode. The F-TENG was composed of two flexible electrodes and one layer of PTFE membrane with their edges sealed up by 3M200C type dual adhesive tape whose width was 2 mm and thickness was 30 μm. The double surfaces of the PTFE membrane were pre-ground with sandpapers, such as P10000 and P400, before assembly.

The structure of the F-TENG network is shown in Figure 7a. The network in this experiment contained two frames made by a 3D printer with polylactic acid (PLA) material. Both frames were free to move to adjust their distance in the horizontal direction. The inner wall of the frame was provided with several equidistant grooves for inserting the flagpole and adjusting the distance of the flag in the vertical direction. Each frame was assembled with six F-TENGs with two flags arranged in the horizontal direction and three in the vertical direction.

### 5.2. Electrical Measurement of the TENG Unit and Network

A programmable electrometer was used to measure the electrical output of the unit and F-TENG. A wind tunnel with the dimensions of 0.25 m (width) × 0.25 m (height) × 1.0 m (length) simulated the wind conditions under which the experiments of the F-TENG unit and the network were performed. The wind speed varied from 2.0 m/s to 8.0 m/s, which was calibrated by a GM8903 type anemometer. At the right end of the wind tunnel, a blower was firmly installed, whose rotating speed was controlled by an inverter in order to adjust the wind speed.

## Figures and Tables

**Figure 1 nanomaterials-12-00721-f001:**
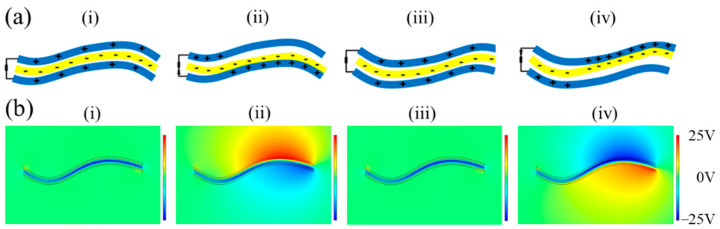
Working principle and simulation of the flag-type triboelectric nanogenerator (F-TENG): (**a**) power generation principle of F-TENG and (**b**) simulation graphic of COMSOL.

**Figure 2 nanomaterials-12-00721-f002:**
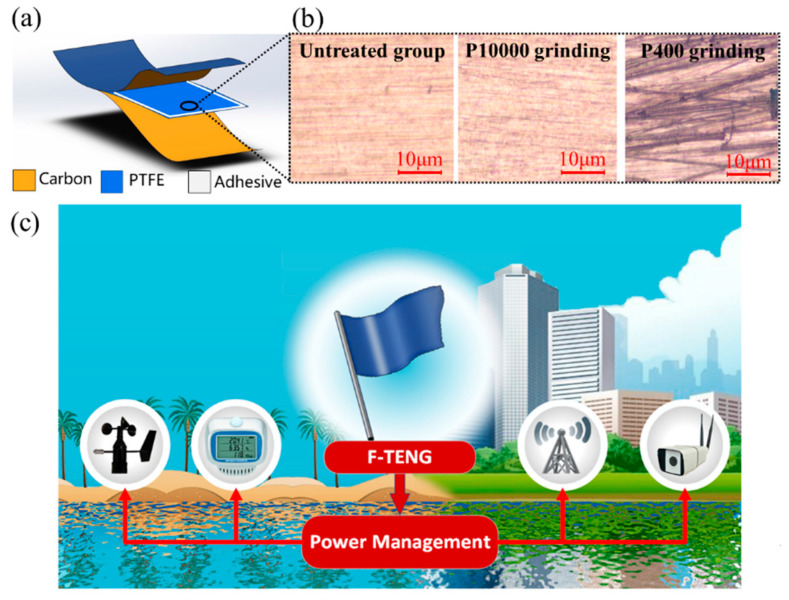
Structure and microstructure of F-TENGs: (**a**) structure and materials of each F-TENG; (**b**) microstructure of different types of polytetrafluoroethylene (PTFE) films after pre-grinding; and (**c**) schematic diagram of F-TENGs applied to wind harvesting and power supply.

**Figure 3 nanomaterials-12-00721-f003:**
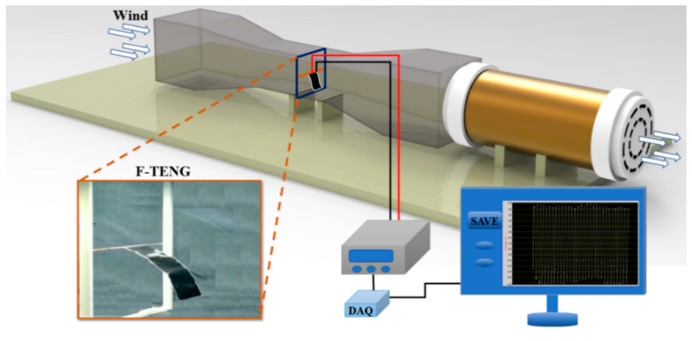
Schematic of the experimental apparatus for assessing F-TENG power generation performance in a wind tunnel.

**Figure 4 nanomaterials-12-00721-f004:**
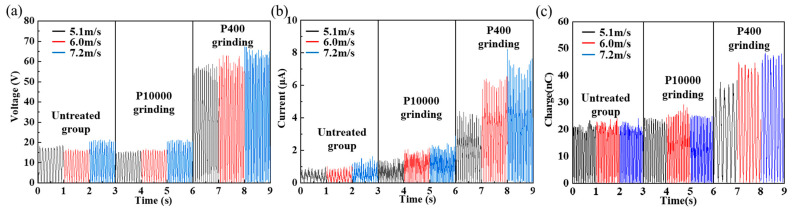
Output of F-TENG: (**a**) output of voltage with the degree of pre-grinding; (**b**) output of current with the degree of pre-grinding; (**c**) output of charge with the degree of pre-grinding.

**Figure 5 nanomaterials-12-00721-f005:**
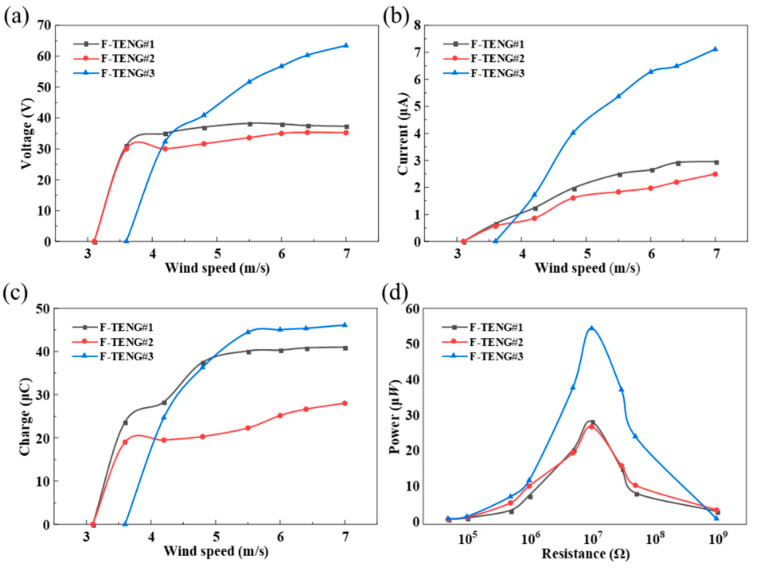
Variation of output of F-TENG#1, #2, and #3 with increasing wind speed: (**a**) voltage output of F-TENGs with different pre-grinding; (**b**) current output of F-TENGs with different pre-grinding; (**c**) charge output of F-TENGs with different pre-grinding; and (**d**) power output of F-TENGs with different pre-grinding.

**Figure 6 nanomaterials-12-00721-f006:**
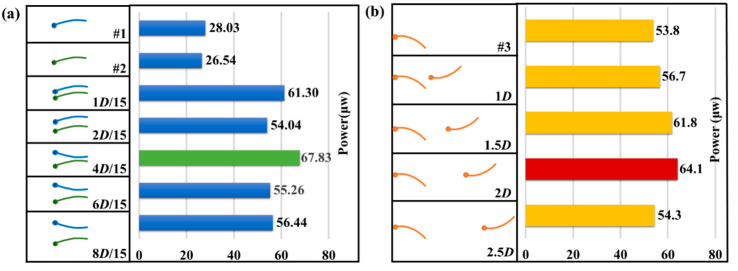
Contrast diagram of the maximum value of total power output of F-TENGs: (**a**) F-TENG #1 and #2 varying with *e*_P_ in vertical parallel mode and (**b**) F-TENG #3 varying with *e*_T_ in horizontal parallel mode.

**Figure 7 nanomaterials-12-00721-f007:**
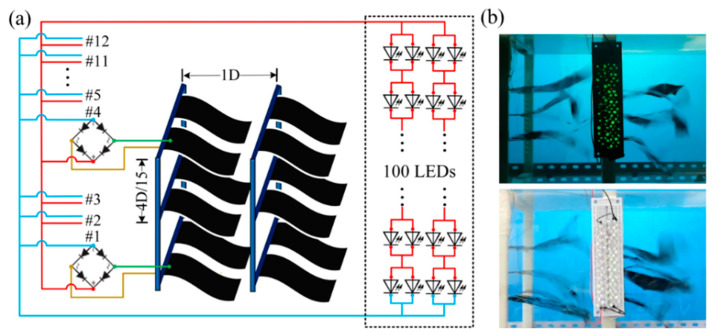
Network and output of F-TENGs: (**a**) schematic diagram of twelve F-TENGs combined for grid power and (**b**) photographs showing the array of twelve F-TENGs lighting up 100 LEDs.

## Data Availability

Not applicable.

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
