# Peer review of "An Array of Flag-Type Triboelectric Nanogenerators for Harvesting Wind Energy"

_nanomaterials, 2022, doi:10.3390/nano12040721_

Round 1
Reviewer 1 Report
In this work the authors experimentally studied flutter type TENG for wind energy harvesting. Importantly they studied interference between multiple flutter TENGs to determine optimal array arrangement in horizontal and vertical directions.
Here are the reviewers comments:
- The proposed flag design is free-standing mode and not contact-separation mode as mentioned in line 87 (Niu, Simiao, et al. Nano Energy 12 (2015): 760-774.)
- In line 113: "Owing to the difference in the sequence of the two electrodes…" should instead be "Owing to the difference in the electron affinity between PTFE and electrode…"
- During pregrinding process of Figure 2, what was the normal pressure/force and horizontal speed of grinding with the sandpaper?
- The word tandem used in section 2.4 and other places is totally misleading. It should be stated as horizontal parallel array and vertical parallel array. Tandem keyword is used for frequency control of TENGs (Kim, Wook, et al. Nano Energy 56 (2019): 307-321.)
- In figure 4 mention current output as peak-to-peak and provide equation for power calculation in Figures 5 and 6
- In figure 6, eT=2D shows highest output, but in Figure 7 and conclusion, eT=1D is written. Further in line 283 eT=6D/15 is written, it should be eP=6D/15.
Author Response
[11/02/2022, Friday]
Dear Editor and Reviewer,
I wish to resubmit an article for publication in Nanomaterials, “An array of flag-type triboelectric nanogenerators for harvesting wind energy.” The manuscript ID is nanomaterials-1592100. The paper was coauthored by Z. Zhao, B. Wei, Y. Wang, X. Huang, B. Li, F. Lin, L. Ma, Q. Zhang, Y. Zou, and F. Yang.
We thank you and the reviewers for your thoughtful suggestions and insights. The manuscript has benefited from these insightful suggestions. I look forward to working with you and the reviewers to move this manuscript closer to publication in Nanomaterials.
The manuscript has been rechecked and the necessary changes have been made in accordance with the reviewers’ suggestions. The responses to all comments have been prepared and attached herewith/given below.
- For Expert Comment #1, #2, and #4, we corrected all the errors as requested.
- For Expert Comment #3, we added the force and horizontal speed of grinding with the sandpaper in line 151.
- For Expert Comment #5, we added the equation for power calculation in line 222.
- For Expert Comment #6, in the demonstration experiment of lighting LED lights, eT =1D showed little difference from eT =2D. In addition, eT =1D is more convenient for shooting, so a demonstration experiment is carried out with eT =1D.
Thank you for your consideration. I look forward to hearing from you.
Sincerely,
Zhiqiang Zhao, Hongchen Pang, Jin Xu, Xinxiang Pan
Maritime College & School of Electronics and Information technology, Guangdong Ocean University
No.1 Haida Road, Mazhang District, Zhanjiang City, Guangdong Province, China
Tel: +86 15907592751
Fax: 0759-2383007
Email: [email protected]

Reviewer 2 Report
The review article (Manuscript ID nanomaterials-1592100) entitled “An array of flag-type triboelectric nanogenerators for harvesting wind energy” by Pan and coworkers explored the wind energy harvesting using flag type triboelectric nanogenerator and this is interesting in scientific community. Abstract and conclusion are satisfactory as it covers the aspects discussed in the manuscript however the reviewer hard to find novelty of the present work discussed in the paper. The authors should address all the comments properly before considering for the publication in the prestigious journal "Nanomaterials".
- The authors claimed they disclosed new method “new method for performance enhancement and the practical application of TENGs” to increase the performance of the TENG. But this type of energy harvester already explored in the energy harvesting community. The authors briefly explain what is the new and novelty of the present work?
- Why the output performances of all the devices were similar at lower wind speeds (4 m/s) and in higher speed F-TENG #3 is drastically increases in higher speed (7 m/s). Explain in detail.
- The authors should put the SEM images of the different devices at different grinding methods. Also calculate the surface charge (responsible for triboelectricity) of the devices at different grinding methods (sandpapers-400, 10000 and untreated).
- Though the output performance of F-TENG #3 device is high but the charging capability of the untreated device is almost similar with F-TENG #3 device (Fig.5C). So, what is the beneficial of the pre-grinding method?
- The reviewer recommends the authors may include some recent energy converting method using piezoelectric/triboelectric energy harvesting technology and storage technology which makes the introduction more interesting and comprehensive. e.g., Science 373, no. 6552, 321-327, 2021, Advanced Materials, 24, 45, 2012, 655-666, Advanced Functional Materials 30 (48), 2004446, 2020, Science 357, 6348, 306-309, 2017, Nanomaterials, 11(12), 3431, ACS Applied Electronic Materials 1 (2), 189-197 etc.
- It will be better to add a table of the output performances compared with the other TENGs related to wind energy harvesting.
Author Response
[15/02/2022, Tuesday]
Dear Editor and Reviewer,
I wish to resubmit an article for publication in Nanomaterials, “An array of flag-type triboelectric nanogenerators for harvesting wind energy.” The manuscript ID is nanomaterials-1592100. The paper was coauthored by Z. Zhao, B. Wei, Y. Wang, X. Huang, B. Li, F. Lin, L. Ma, Q. Zhang, Y. Zou, and F. Yang.
We thank you and the reviewers for your thoughtful suggestions and insights. The manuscript has benefited from these insightful suggestions. I look forward to working with you and the reviewers to move this manuscript closer to publication in Nanomaterials.
The manuscript has been rechecked and the necessary changes have been made in accordance with the reviewers’ suggestions. The responses to all comments have been prepared and attached herewith/given below.
- For Expert Comment #1: As described in the article, we proposed a method of pregrinding PTFE electret film, which improved the power generation performance of F-TENG. At the same time, for the practical application of the array problem, we have carried out theoretical analysis and experimental research. The research ideas and direction are provided for the performance improvement and the practical application research of TENG.
- For Expert Comment #2: At lower wind speed (4-5 m/s), the flag flutter frequency is low, and the contact area between PTFE film and conductive carbon film is small, so the power output is relatively similar. With the increase of wind speed, the contact area between PTFE film and conductive carbon film increases, and the power output shows obvious difference. When the wind speed is close to 7 m/s, the amount of transferred charge tends to be constant. However, as the wind speed increases, the frequency and amplitude of flag flutter increase, the voltage and current output of F-TENG #3 show an obvious trend of increase.
- For Expert Comment #3: Please understand that our laboratory is not equipped with SEM capability. However, the pictures taken by industrial electron microscopy can basically reflect the surface morphology of PTFE film after grinding. Please refer to Figure 2b. As requested, we have inserted the surface charge data curve into Figure 4c.
- For Expert Comment #4: F-TENG #1, #2, #3 are all pre-grinding devices. However, F-TENG #1 and #2 are light pre-grinding devices treated with P10000 sandpaper. Manual pre-grinding process was used in the experiment, and the output performance of each flag was slightly different. The overall results show that pre-grinding can effectively improve the output performance of F-TENG.
- For Expert Comment #5: Thank you for your comprehensive analysis of the introduction. Several papers recommended by you have played an important role in guiding the introduction. At the same time, it also opens up the horizon of our authors to learn more relevant knowledge. Several references related to this paper have been cited in this paper. Thank you again for your guidance.
- For Expert Comment #6: Table 1 below is the summary of output performance and working wind speed of various wind energy harvesting TENGs. The F-TENG processed by our experiment can obviously show excellent output performance. However, we think it is appropriate to use this table as a supplementary document, and it may not be appropriate to list it in the body of the paper. We look forward to further comments from editors and reviewers.
Table 1. A summary of output performance and working wind speed of various wind energy harvesting TENGs
Device |
Power density ( ) |
U (m/s) |
Ref. |
Wind energy harvesting TENG |
0.0012 |
10 |
[1] |
F-TENG |
0.0011 |
12.1 |
[2] |
Flag-type TENG |
0.0408 |
7.5 |
[3] |
F-TENG #3 |
0.0598 |
7.0 |
present |
Hybrid nanogenerator |
0.0616 |
10.2 |
[4] |
Flow-driven TENG |
0.25 |
7.6 |
[5] |
AF-TENG |
0.47 |
22 |
[6] |
TENG for wind energy harvesting |
0.782 |
30 |
[7] |
ATNG |
1 |
20 |
[8] |
Wind-driven TENG |
2.2 |
18 |
[9] |
Elasto-aerodynamics-driven TENG |
9 |
15 |
[10] |
[1] Y. Yang, G. Zhu, H. Zhang, J. Chen, X. Zhong, Z.-H. Lin, Y. Su, P. Bai, X. Wen, Z.L. Wang, ACS Nano, 7 (2013) 9461-9468.
[2] Y. Su, G. Xie, X. Tao, H. Zhang, Z. Ye, Q. Jing, H. Tai, X. Du, Y. Jiang, J. Phys. D Appl. Phys., 49 (2016) 215601.
[3] Y. Wang, E. Yang, T. Chen, J. Wang, Z. Hu, J. Mi, X. Pan, M. Xu, Nano Energy ,78 (2020) 105279.
[4] M. Ma, Z. Zhang, Q. Liao, G. Zhang, F. Gao, X. Zhao, Q. Zhang, X. Xun, Z. Zhang, Y. Zhang, Nano Energy, 39 (2017) 524-531.
[5] S. Wang, X. Mu, Y. Yang, C. Sun, A.Y. Gu, Z.L. Wang, Adv Mater, 27 (2015) 240-248.
[6] M. Xu, Y. Wang, S.L. Zhang, W. Ding, J. Cheng, X. He, P. Zhang, Z. Wang, X. Pan, Z.L. Wang, Extrem. Mech. Lett., 15 (2017) 122-129.
[7] Y. Wang, J. Wang, X. Xiao, S. Wang, T.K. Phan, J. Dong, J. Mi, X. Pan, H. Wang, M. Xu, Nano Energy, 73 (2020) 104736.
[8] H. Guo, X. He, J. Zhong, Q. Zhong, Q. Leng, C. Hu, J. Chen, L. Tian, Y. Xi, J. Zhou, J. Mater. Chem. A, 2 (2014) 2079-2087.
[9] H. Zheng, Y. Zi, X. He, H. Guo, Y.C. Lai, J. Wang, S.L. Zhang, C. Wu, G. Cheng, Z.L. Wang, ACS Appl Mater Interfaces, 10 (2018) 14708-14715.
[10] S. Wang, X. Mu, X. Wang, A.Y. Gu, Z.L. Wang, Y. Yang, ACS Nano, 9 (2015) 9554-9563.
Thank you for your consideration. I look forward to hearing from you.
Sincerely,
Zhiqiang Zhao, Hongchen Pang, Jin Xu, Xinxiang Pan
Maritime College & School of Electronics and Information technology, Guangdong Ocean University
No.1 Haida Road, Mazhang District, Zhanjiang City, Guangdong Province, China
Tel: +86 15907592751
Fax: 0759-2383007
Email: [email protected]
